environmental science

food waste, anaerobic digestion, waste bioconversion, yeast, biogas, propionic acid inhibition

**Author for correspondence:**
Chuanfu Wu
e-mail: wuchuanfu83@gmail.com

# Effect of yeast addition on the biogas production performance of a food waste anaerobic digestion system

Ming Gao[1,2,†], Shuang Zhang[1,†], Xinxin Ma[1], Weijie Guan[1], Na Song[3], Qunhui Wang[1,2] and Chuanfu Wu[1,2]

[1]Department of Environmental Science and Engineering, School of Energy and Environmental Engineering, and [2]Beijing Key Laboratory on Resource-oriented Treatment of Industrial Pollutants, University of Science and Technology Beijing, Beijing 100083, People's Republic of China
[3]Department of Environmental Engineering, Tianjin College, University of Science and Technology Beijing, Tianjin 301830, People's Republic of China

MG, 0000-0001-8287-9298; CW, 0000-0002-5404-0783

Food waste contains numerous easily degradable components, and anaerobic digestion is prone to acidification and instability. This work aimed to investigate the effect of adding yeast on biogas production performance, when substrate is added after biogas production is reduced. The results showed that the daily biogas production increased 520 and 550 ml by adding 2.0% (volatile solids; VS) of activated yeast on the 12th and 37th day of anaerobic digestion, respectively, and the gas production was relatively stable. In the control group without yeast, the increase of gas production was significantly reduced. After the second addition of substrate and yeast, biogas production only increased 60 ml compared with that before the addition. After fermentation, the biogas production of yeast group also increased by 33.2% compared with the control group. Results of the analysis of indicators, such as volatile organic acids, alkalinity and propionic acid, showed that the stability of the anaerobic digestion system of the yeast group was higher. Thus, the yeast group is highly likely to recover normal gas production when the biogas production is reduced, and substrate is added. The results provide a reference for experiments on the industrialization of continuous anaerobic digestion to take tolerable measures when the organic load of the feed fluctuates dramatically.

[†]Ming Gao and Shuang Zhang contributed equally to this work.

# 1. Introduction

Food waste (FW) is one of the most important components in municipal solid waste. The proportion is 30–60% in China, while it is 23% in Japan, 15–25% in Europe, and only 12% in the United States. FW has the characteristics of high moisture content, high organic content and easy degradability [1,2]. Improper handling of FW would cause problems of food safety and environmental pollution. At present, FW is mostly processed through landfill and incineration as final disposal. However, these treatment approaches may cause secondary pollution to the environment and increase in processing costs [3,4]. FW possesses rich nutrition, good biodegradability and high methanogenic potential, which is considered as a good substrate for methane fermentation. Anaerobic digestion (AD) is more appropriate for FW treatment, because complex organic components could be degraded. On the other hand, biogas produced from AD is easily separated and used in applications [5]. Further, AD has the advantages of simple process flow, high reactor efficiency, low pollution load, high economic benefits and even the highest potential for FW recycling.

FW is regarded as a high-quality raw material for AD [6,7]. Through the AD process, organic matter in FW can be converted into methane, carbon dioxide and some other substances. As the main product, methane can be used as an energy substance. However, the application of AD is often limited by its long lag time and low methane production rate [8]. FW is always characterized with low C/N ratio [9]. Thus, the AD process with FW often results in the accumulation of volatile fatty acids (VFAs) in the reaction system, due to the imbalanced nutrition in the substrates [10,11]. Further, the acidification reduces methane yield and destroys the stability of the digestive system, leading to imbalance.

Previous studies have identified several methods to enhance methane fermentation efficiency using FW substrate pretreatments, including mechanical, thermal, chemical and biological pretreatments [12,13]. Mechanical pretreatment enhances anaerobic process by increasing specific surface area. Li *et al.* [14] studied the effect of heat pretreatment on the degradation of organics in FW, found that heat pretreatment increased the stagnation period of protein degradation (35–65%) and the cumulative gas production also increased linearly. However, these methods require costly facilities and treatments. As an important part of biological pretreatment, ethanol pre-fermentation has become a popular research topic in recent years. Wu *et al.* found that ethanol pre-fermentation effectively alleviated the inhibition of acidification, greatly reduced the lag period and significantly stimulated the growth of methanogens [15]. Although adding sodium hydroxide or bicarbonate to the AD system is the most direct and effective approach to control the pH of the system [16,17], these methods lead to excessive sodium ion accumulation and AD system inhibition. Zero-valent iron is a new type of additive for AD system, which can restore excessive acidification, relieve low pH by stimulating the metabolic performance of microorganisms and reduce the redox potential [18]. Kong *et al.* [19] found that the addition of zero-valent iron can alleviate excessive acidification, which is beneficial for the conversion of dominant microbial species from methanogens to methanogens and methanosarcina, thereby alleviating the accumulation of non-acetic acid VFAs. However, the approach of supplementing zero-valent iron would take a long time to relieve acidification and restore methane production, which caused low efficiency.

Yeast has the characteristics of osmotic pressure resistance, acid resistance and high metabolic efficiency, which is widely used in the treatment of high-concentration organic wastewater, including the treatment of toxic and refractory pollutant-containing wastewater [20]. It has been proved that yeast can accelerate reaction rates in the hydrolysis stage of organic matter and promote the hydrolysis of FW [21]. FW contains high enough levels of proteins, amino acids and trace elements from the hydrolysis and breakdown of food-stuffs, and can provide sufficient nutrition for the growth of yeast [22]. In the process of anaerobic digestion of food waste, yeast can convert the degradable organic matter of substrates into neutral ethanol instead of acid propionic acid or butyric acid, which reduces the load of VFAs in the system [6,23]. Through the addition of yeast to the FW substrate for ethanol pre-fermentation before AD, a large amount of ethanol is produced, and ethanol is gradually converted into acetic acid, which in turn can be easily used by methanogens, thereby increasing methane production, reducing the effect of acidification on the system and enhancing the stability of AD [15,24]. However, there are few studies on adding yeast directly to the anaerobic digestion system of FW. The aim of this study was to investigate the effect of yeast addition on the AD system of FW. In this work, activated yeast was added to the fermentation broth of anaerobic fermentation of FW. On the other hand, a control group without yeast addition was established. The daily gas production, cumulative gas production, pH, alkalinity, VFAs and other indicators of these two systems were compared and analysed. Here, the effect of yeast addition on the methane production from FW in AD system was conducted.

**Table 1.** Characteristics of food waste and inoculum sludge.

| parameters | food waste | inoculum sludge |
|---|---|---|
| total solids (%) | $25.7 \pm 0.03$ | $12.2 \pm 0.01$ |
| volatile solids (%) | $24.0 \pm 0.03$ | $6.89 \pm 0.03$ |
| pH | 4.70 | 8.84 |
| carbohydrate[a] (%) | $45.0 \pm 0.03$ | $68.1 \pm 0.03$ |
| protein[a] (%) | $14.8 \pm 0.07$ | $14.5 \pm 0.04$ |
| fat[a] (%) | $31.8 \pm 0.5$ | —[b] |
| C[a] (%) | $49.1 \pm 0.04$ | $27.5 \pm 0.09$ |
| N[a] (%) | $2.10 \pm 0.14$ | $2.50 \pm 0.26$ |
| C/N | $23.5 \pm 1.6$ | $11.1 \pm 1.2$ |
| H[a] (%) | $7.23 \pm 0.15$ | $5.67 \pm 0.22$ |
| O[a] (%) | $30.2 \pm 0.17$ | $25.8 \pm 0.34$ |

[a]refers to dry basis.
[b]not detected.

# 2. Material and methods

## 2.1. Feedstock and inoculums

The FW used in the experiments were collected from the student canteen of the University of Science and Technology Beijing. Large bones and pericarps of FW were first removed. The remaining waste was shredded and then stored at 20°C. The inoculated sludge was taken from a rural biogas station in Beijing's Pinggu District and used after acclimation. Table 1 shows the physical and chemical properties of FW and sludge. Active dry yeast was produced by Angel Yeast Co., Ltd, Hubei, China, which is used as an additive to the anaerobic digestion system. Yeast contains 40–60% protein, 25–35% carbohydrate, 4–7% fat and 6–15% nucleic acid. The glucose was produced by Aladdin Co., Ltd, Shanghai, China.

## 2.2. Experimental methods

In this study, two sets of experiments were conducted in batch mode. The AD reactions were carried out in 500 ml anaerobic bottles with 400 ml working volume. Two sets of AD experiments were designed, namely, a yeast group with activated yeast and a control group without yeast. The addition ratio of FW and sludge in the two groups of experiments was 1.25 : 1 (based on volatile solids; VS). In the yeast group, 2.0% (calculated as VS) of activated *Saccharomyces cerevisiae* was added when fermentation gas production had drastically reduced (day 12 of fermentation) to investigate the effect of yeast on reducing the load of VFAs, improving the system's stability and methane production. The method for the yeast addition and activation has been mentioned in previous research [15]. Given that 2.0% (calculated as VS) glucose was added during yeast activation, glucose of the same quality was also added to the control group to eliminate the effect of glucose on the methanogenesis performance of the FW anaerobic digestion system. In order to investigate whether the reduction of gas production to zero in the later stage of anaerobic fermentation is due to the depletion of substrate or the deactivation of methanogens, yeast and glucose were added for the second time (no biogas generation, 37 days). The experimental flow chart is shown in figure 1. The two groups of reactors were filled with nitrogen to discharge excess air and then sealed at 37°C and 60 r.p.m. for experiments. The experiments lasted for 50 days. Sampling was withdrawn and various parameters were determined at regular intervals. Biogas production was recorded daily.

## 2.3. Analytical procedures

The contents of total solids (TS) and volatile solids (VS) of the solid samples were determined in accordance with the method determined by the State Environmental Protection Administration (2002). PHS-3C digital acidity (INESA Scientific Instrument Co., Ltd, Shanghai, China) was used to measure pH. Alkalinity was measured through bromocresol green–methyl red indicator titration method. Fat content was measured

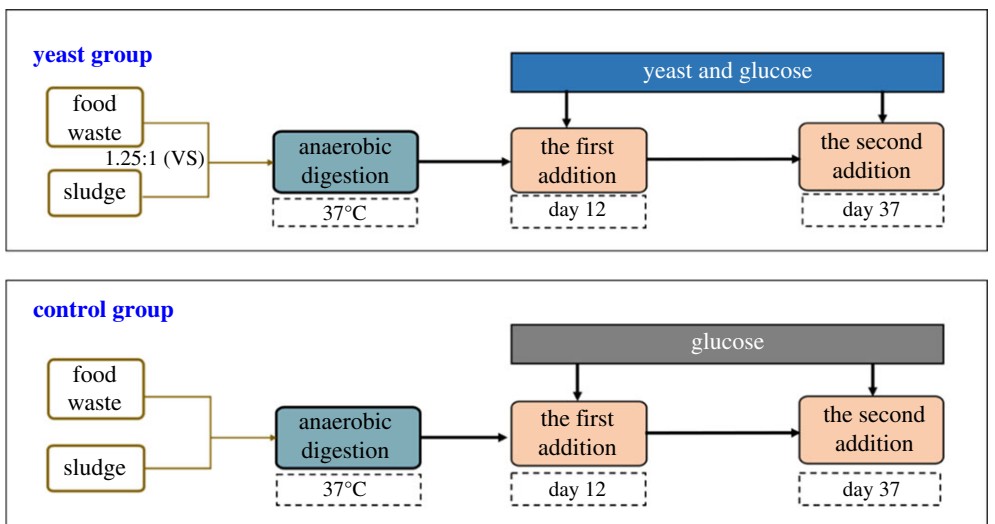

**Figure 1.** Anaerobic digestion flow chart.

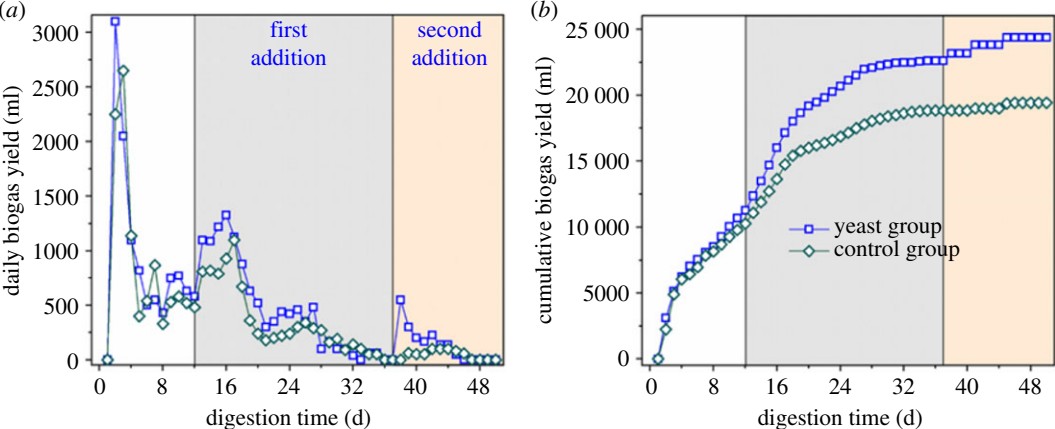

**Figure 2.** Daily biogas production (*a*) and cumulative biogas production (*b*) by the yeast and control groups.

using the national standard GB 5009.6–2016. Protein content was quantified using K9840 automatic nitrogen analyser (Shandong Haineng Scientific Instrument Co., Ltd, Jinan, China). The VFAs composition was measured using the following method. First, the samples were centrifuged at $12\,000 \times g$ for 10 min. Second, the supernatant was diluted and filtered through a 0.22 µm microporous membrane. Then, the samples were determined on a Shimadzu CP3800 gas chromatograph equipped with a DB-FFAP capillary column (30 m × 0.53 mm × 0.5 µm, Agilent Technologies Co., Ltd) and a flame ionization detector. The biogas was collected by the drainage method with alkaline.

# 3. Results and discussion

## 3.1. Comparison between the performances of the two groups in anaerobic digestion biogas production

Experiments were performed on the yeast and control groups in accordance with the method described in §2.2. The experimental conditions for the two groups were exactly the same, except that the yeast group was treated with 2.0% more yeast (in VS) than the control group on days 12 and 37. The time course of daily biogas production and cumulative biogas production of two groups within anaerobic fermentation were recorded (figure 2).

Figure 2*a* indicates that the daily output of biogas in both groups was decreased considerably (2–12 days). However, when the same amount of glucose was added to the two groups on day 12 and after yeast was added to the yeast group, daily biogas production by both groups drastically increased. The

amounts of biogas in the yeast and control groups on day 13 increased by 520 and 330 ml, respectively, relative to those in the same groups before day 12. During days 13 to 37, gas production by the yeast group was 115.6 ml day$^{-1}$ higher than that in the control group on average.

Cumulative biogas production by the two experimental groups is shown in figure 2b. In the absence of yeast (1–12 days), the cumulative biogas production trends of the two groups remained the same, and gas production did not significantly differ. After the first addition, the cumulative gas production of the yeast group was 26.5% higher than that of the control group. After the second addition, the cumulative biogas production of the yeast group was 34.6% higher than that of the control group and reached 417 ml g$^{-1}$ VS. At the end of fermentation (day 50), the cumulative biogas production of the yeast group was 449.2 ml g$^{-1}$ VS and that of the control group was 333.8 ml g$^{-1}$ VS. Under standard conditions, 1 g of protein (calculated as VS) can produce 496 ml of biogas in anaerobic fermentation [25]. The calculation formula for protein biogas production is shown in equation (3.1). The average protein content in yeast was 50%, and the amount of biogas produced by adding 1 g yeast itself was 248 ml. Therefore, deducting 4.6 ml g$^{-1}$ VS of the biogas produced by the yeast itself in the yeast group revealed that biogas production by the yeast group had increased by 33.2% compared with that by the control group. The addition of activated yeast can drastically promote the anaerobic digestion of methanogenesis in FW. In other study, yeast was added to the substrate for ethanol pre-fermentation in the sequencing batch methane fermentation of food waste. The results showed that methane production in the ethanol pre-fermentation group (254 ml g$^{-1}$ VS) was higher than in the control group (35 ml g$^{-1}$ VS) [24]. It was consistent with the results of this study.

$$C_5H_7NO_2 + 3H_2O \rightarrow 2.5CH_4 + 2.5CO_2 + NH_3. \tag{3.1}$$

This effect may be attributed to the reduction in the production of VFAs by the addition of activated yeast. Because of the glycolytic metabolic pathway in *S. cerevisiae*, the end products are mainly ethanol and acetic acid after yeast addition [26]. Moreover, the acidification in AD system is considered as the problematic inhibition of biogas production. The organic matter in FW was mainly converted into ethanol by adding yeast, instead of VFAs. The ethanol could be gradually converted into acetic acid, which can be easily used by methanogens, thereby increasing methane production and enabling the stable operation of anaerobic fermentation [4]. And the addition of yeast can improve the relative abundance of methane-producing bacteria in anaerobic digestion system [24].

## 3.2. Comparison of the pH, alkalinity and total volatile fatty acid of the two groups during anaerobic digestion

The pH, alkalinity (TA), total volatile fatty acid (TVFA) and TVFA/TA are crucial indices that reflect the stability of anaerobic fermentation [27]. The changes in pH, alkalinity, TVFA concentration and TVFA/TA of two experimental groups with anaerobic fermentation time are shown in figure 3.

Controlling the pH value can increase the hydrolysis and acidification rate of anaerobic fermentation, and inappropriate pH value can inhibit the fermentation of organic matter [28]. As shown in figure 3a, the pH values of the two experimental groups gradually decreased but were never less than 6.5. Therefore, acidification did not occur during anaerobic fermentation. This phenomenon also explains why the daily output of biogas in the early stage of fermentation in figure 2a did not decrease to zero, even if it decreased. After 8 days of anaerobic fermentation, the pH of the two groups of experiments rapidly increased probably because of the buffering effect of the system itself. After the first addition of yeast and glucose, the pH value of the yeast group did not significantly differ from that of the control group, which was above 7.5. According to current research reports, due to the differences in the nature of the additives, the pH of a methane fermentation system that can maintain stable operation is in the range of 6.5–8.2 [29]. The pH value also plays a vital role in regulating the activities of microorganisms. The optimum pH range of acid-producing bacteria is in the range of 4.0–8.5 and the optimum pH range of methanogenic bacteria is in the range of 6.5–7.2 [30].

Given that the alkalinity of the fermentation broth (attributed to HCO$_3^-$, CO$_3^{2-}$ and other substances that can accept proton H$^+$) can neutralize a certain amount of acidic substances [31], alkalinity decreases drastically with increasing TVFA. As shown in figure 3b, the changes in the alkalinity of the two groups were the same, but the alkalinity of the yeast group was slightly higher than that of the control group. For example, that of the former was 1500.0 mg l$^{-1}$, and that of the latter was 1122.7 mg l$^{-1}$ on the 37th day. The addition of yeast may promote the hydrolysis of organics to produce neutral ethanol, reduce the amount of VFAs and reduce the consumption of alkalinity in the system.

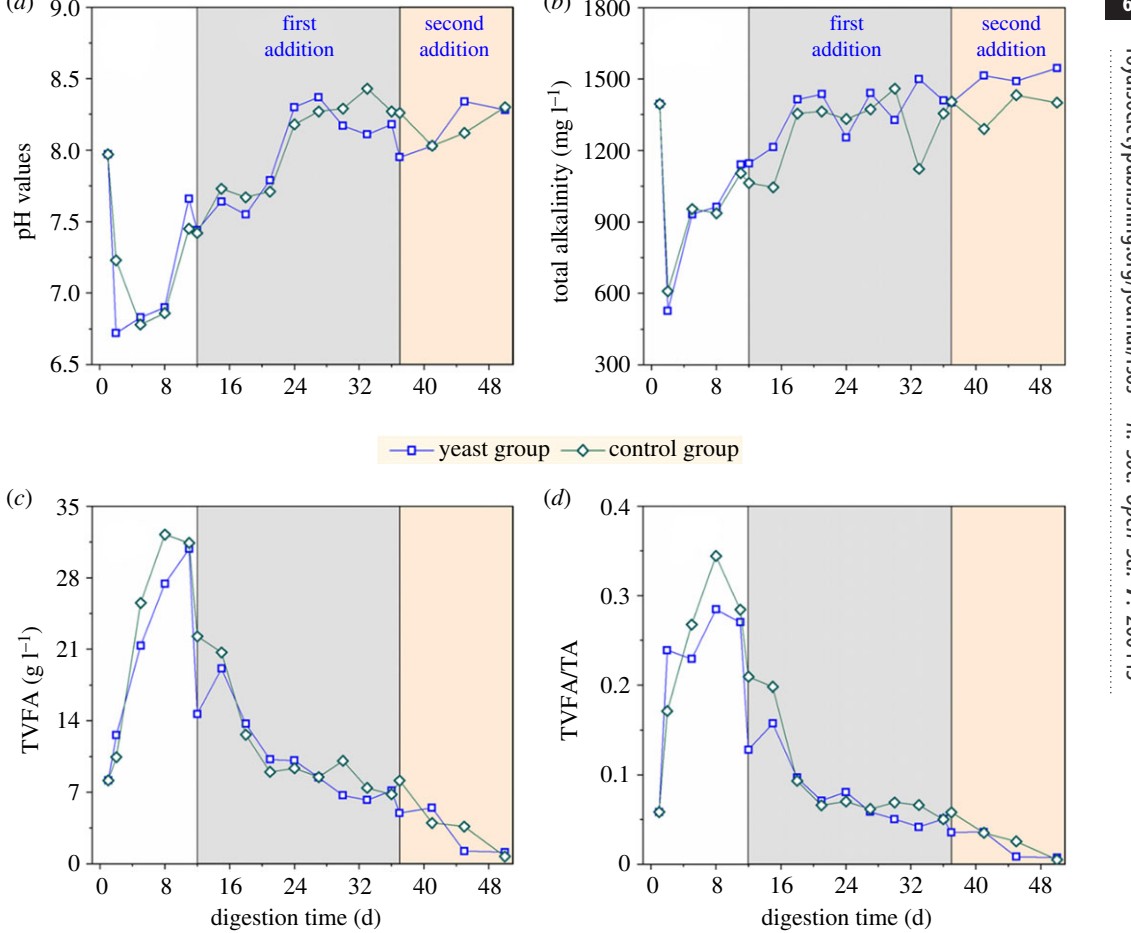

**Figure 3.** Time course of pH (*a*), alkalinity (*b*), TVFA concentration (*c*) and TVFA/TA (*d*) of the yeast group and the control group during anaerobic digestion.

VFAs are important intermediates produced through the degradation of organic matter during anaerobic fermentation, which are the basis for methanogens to produce methane and are important indicators of the metabolic activity of anaerobic microorganisms [32]. Excessive VFAs inhibit the system. As shown in figure 3*c*, the TVFA concentration of the two groups increased rapidly at the start of the fermentation. With the increase of fermentation time, the TVFA concentration of the two groups continued to decrease in line with the increase in pH and alkalinity illustrated in figure 3*a,b*.

Figure 3*d* shows the variation in TVFA concentration to TA (TVFA/TA) values with fermentation time in the two groups of experiments. After the addition of activated yeast on day 12 of fermentation, the alkalinity of the system increased rapidly (figure 3*b*). The TVFA/TA ratio continued to decrease, and the yeast system began to produce a large amount of gas. The TVFA/TA value of the yeast group was lower than that of the control group, indicating that the addition of yeast to the anaerobic digestion system can adjust alkalinity and reduce TVFA concentration, thereby improving the stability of anaerobic digestion. The TVFA/TA values of the two groups of experiments never exceeded 0.4. This result also proved that the system was not acidified, and the reduction in gas production before the addition of yeast or glucose may be caused by substrate exhaustion. Similar findings have been found in previous studies. The ratio of TVFA/TA can be used as an early warning of digestive system imbalance. TVFA/TA shows the ratio between compounds that causes a decrease in pH and compounds that maintain alkalinity in the system. This index sensitively reflects the ability of the anaerobic digestion system to withstand acidification. When TVFA/TA exceeds 0.4, the acidification of the anaerobic system is about to be destabilized. When TVFA/TA exceeds 0.6, the acidification of the anaerobic system will be completely imbalanced [23]. Comparing the two groups' parameters, such as pH, alkalinity, TVFA concentration and TVFA/TA ratio, revealed that these parameters followed the same trend. However, the alkalinity of the yeast group was slightly higher

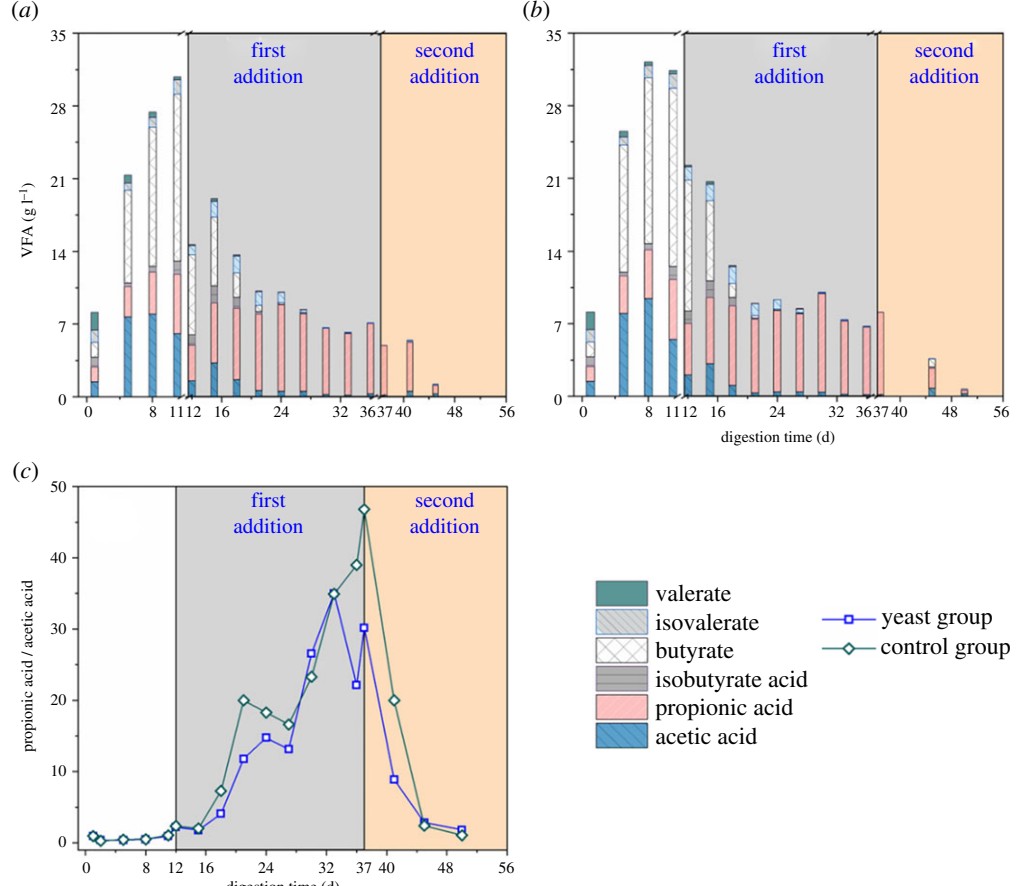

**Figure 4.** The variations of VFAs compositions (*a,b*) and propionic acid/acetic acid (*c*) in the yeast group and the control group.

than that of the control group, the TVFA and TVFA/TA of the yeast group were slightly lower, thereby indicating that the addition of activated yeast improved the stability of anaerobic digestion.

## 3.3. Composition of volatile organic acids generation of the two groups during anaerobic digestion

The main VFAs components detected in the two groups were acetic, propionic and butyric acids (figure 4*a,b*). Valeric acid, isovaleric acid and isobutyric acid were also detected at low concentrations. The acetic acid concentration of the yeast group reached a maximum of 7.97 g l$^{-1}$ on day 8 and then began to decrease with the consumption of butyric acid. From days 11 to 15, the concentration of butyric acid decreased rapidly. Daily biogas production also increased (figure 2*a*). Propionic acid gradually accumulated with the progression of fermentation, thereby propionic acid-type fermentation was considered to be established. In anaerobic fermentation, VFAs containing more than two carbon chains cannot be directly used as a substrate by methanogens and thus are easily accumulated during fermentation [3]. Propionic acid is a common short-chain fatty acid. From the perspective of metabolism, propionic acid is usually converted into acetic acid and hydrogen. However, the process of propionic acid and butyric acid forming acetic acid is endothermic reaction, which is difficult to be carried out in thermodynamics, and the process of propionic acid converting into acetic acid is the most difficult. It was suggested that propionic acid is a disadvantageous substrate for microorganisms [19]. Therefore, when propionic acid-type fermentation has occurred, the utilization of organic acids in the methanogenic phase is inhibited and acids accumulation is promoted, which adversely affects methanogenesis. It was found that high temperature and alkaline pH are favourable for propionic acid generation [33]. On the other hand, the yield of VFAs, especially propionic acid, were also suggested to be enhanced by adjusting the pH level. The key enzyme activity associated with propionic acid formation was the highest, when the system pH was set at 8.0 [3]. After 12 days of fermentation, the propionic acid concentration of the yeast group was lower than that of the control

group, indicating that the addition of yeast can play a certain preventive role when propionic acid fermentation has occurred in the anaerobic fermentation system.

The ratio of the concentration of propionic acid to acetic acid (PC/AC) can be used as an indicator to verify the presence of inhibitory propionic acidification in the digestive system and can predict the acidification of the system with high accuracy. High PC/AC value is associated with the poor stability of anaerobic fermentation. The changes in the PC/AC values of the yeast group and the control group are shown in figure 4c. The PC/AC value increased continuously as anaerobic fermentation progressed because a large amount of acetic acid was used in the early stage of fermentation, which reduced acetic acid content. At the same time, propionic acid accumulated because it cannot be used directly. The lower PC/AC value of the yeast group than that of the control group indicated that the addition of yeast can be used to prevent the formation of propionic acid. The research results of Zhao *et al*. showed that the addition of yeast can increase the production of ethanol and reduce the production of lactic, propionic and butyric acids [34]. In addition, the product ethanol can be used as a slow-release matrix that continuously releases acetic acid into the anaerobic system to alleviate the acidification of the system and increase methane yield.

# 4. Conclusion

The effect of yeast addition on AD of FW was investigated in this study. The results showed that the addition of yeast can restore and promote the biogas production in AD system. Moreover, AD with yeast addition exhibited a high VFAs consumption rate and low propionic acid concentration, which prevented the excessive acidification phenomenon. By adding yeast, FW was converted into ethanol as a slow-release substrate, instead of VFAs. The ethanol could be gradually converted into acetic acid, which can be easily used by methanogens, thereby increasing methane production and enabling the stable operation. Therefore, yeast addition was suggested as a feasible approach to maintain a stable AD system.

Data accessibility. Data available from the Dryad Digital Repository: https://doi.org/10.5061/dryad.9s4mw6md0 [35].
Authors' contributions. M.G. and S.Z. contributed in drafting the manuscript, planning and performing the experiments. X.M., W.G. and N.S. contributed in analysing the data. Q.W. contributed in reviewing and revising the manuscript. C.W. contributed in supervision and editing. All authors read and approved the final manuscript to be published.
Competing interests. The authors declare that they have no known competing financial interests or personal relationships that could have appeared to influence the work reported in this paper.
Funding. This work was supported by the Beijing Natural Science Foundation Program (8192028), the National Key R&D Program of China (2018YFC1900903 and 2018YFC1900904) and the Foundation of the Committee on Science and Technology of Tianjin (18YFHBZC00020).
Acknowledgements. The authors gratefully acknowledge support for this study from the National Environmental and Energy Base for International Science & Technology Cooperation.

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
