## [Reviewer comments · Royal Society Open Science]

Review History

RSOS-200443.R0 (Original submission)

Review form: Reviewer 1

Is the manuscript scientifically sound in its present form?

Yes

Are the interpretations and conclusions justified by the results?

Yes

Is the language acceptable?

Yes

Do you have any ethical concerns with this paper?

No

Have you any concerns about statistical analyses in this paper?

No

Recommendation?

Major revision is needed (please make suggestions in comments)

Comments to the Author(s)

In this paper, effects of yeast addition on the biogas production performance of a food waste (FW) anaerobic digestion system were comprehensively investigated. Generally, the topic and the data analysis are significant, the results obtained would make a sound contribution to the fields of sludge treatment and anaerobic digestion. However, in the reviewer's opinion, there are some major and minor shortcomings needed further attention via a major revision, with the detail as follows:

1. What is the role of yeast in the anaerobic digestion? I find there are a contradictory statement about it in the introduction. Firstly, the author stated " the AD process with FW often results in the accumulation of volatile organic acids (VFAs) in the reaction system and the acidification reduces methane yield and destroys the stability of the digestive system" in Page 3, Line 17-20. Then in Page 4, Line 10-12, the authors stated " The addition of yeasts during anaerobic digestion can increase the acid production rate of anaerobic digestion and shorten the time of anaerobic digestion". Both statements obviously are contradictory. What exactly is the mechanism for enhanced digestion of FW led by yeast? It should be confirmed.
2. In this study, the yeast was added in twice. What is the reason? and how to chose the added time? Meanwhile, the characteristic of the yeast *Saccharomyces cerevisiae* should be provided.
3. Except for the yeast, some other pretreatment technologies are benefit to the anaerobic digestion of FW and they should be summarized and compared in the introduction. Some recent references maybe helpful, such as Reviews in Environmental Science and Bio-technology, 2019, 18(4):771-793(<https://doi.org/10.1007/s11157-019-09515-y>); Environmental Science and Pollution Research, 2019, 26(14): 13984- 13998 (<https://doi.org/10.1007/s11356-019-04798-8>)
4. In this study, the substrate also was added secondly. Why?
There are only one equation, so using 3.1 to label is improper.
5. Numerous grammar errors distributed across the paper, resulting in poor readability. I think the article in its current state is unable to be accepted unless the article will be re-written by some of the co-authors who are skillful in English.
for example:
Page 2, Line 4 " ...under decreased substrate conditions after biogas production was reduced was studied." ???
Page 2, Line 6-7, what do you mean " the recovery of gas production"?
Page 4, Line 8 " the effect and performance of direct yeast addition"??
Page 4, Line 19-20 " FW and sludge anaerobic fermentation."???

Review form: Reviewer 2

Is the manuscript scientifically sound in its present form?

Yes

Are the interpretations and conclusions justified by the results?

Yes

Is the language acceptable?

No

Do you have any ethical concerns with this paper?

No

Have you any concerns about statistical analyses in this paper?

No

Recommendation?

Accept with minor revision (please list in comments)

Comments to the Author(s)

The authors must correct the grammar and choose proper words to make the manuscript more readable and ready for publication

Review form: Reviewer 3

Is the manuscript scientifically sound in its present form?

No

Are the interpretations and conclusions justified by the results?

No

Is the language acceptable?

No

Do you have any ethical concerns with this paper?

No

Have you any concerns about statistical analyses in this paper?

No

Recommendation?

Major revision is needed (please make suggestions in comments)

Comments to the Author(s)

Please find the attached file (Appendix A) for review comments to enable you improve the quality of your manuscript.

Decision letter (RSOS-200443.R0)

Dear Dr Gao,

The editors assigned to your paper ("Effect of yeast addition on the biogas production performance of a food waste anaerobic digestion system") have now received comments from reviewers. We would like you to revise your paper in accordance with the referee and Associate Editor suggestions which can be found below (not including confidential reports to the Editor). Please note this decision does not guarantee eventual acceptance.

Please submit a copy of your revised paper before 28-Jun-2020. Please note that the revision deadline will expire at 00.00am on this date. If we do not hear from you within this time then it will be assumed that the paper has been withdrawn. In exceptional circumstances, extensions may be possible if agreed with the Editorial Office in advance. We do not allow multiple rounds of revision so we urge you to make every effort to fully address all of the comments at this stage. If deemed necessary by the Editors, your manuscript will be sent back to one or more of the

original reviewers for assessment. If the original reviewers are not available, we may invite new reviewers.

- Data accessibility

If you wish to submit your supporting data or code to Dryad (<http://datadryad.org/>), or modify your current submission to dryad, please use the following link:
<http://datadryad.org/submit?journalID=RSOS&manu=RSOS-200443>

- Competing interests

- Authors' contributions

AB carried out the molecular lab work, participated in data analysis, carried out sequence alignments, participated in the design of the study and drafted the manuscript; CD carried out

the statistical analyses; EF collected field data; GH conceived of the study, designed the study, coordinated the study and helped draft the manuscript. All authors gave final approval for publication.

- Acknowledgements

- Funding statement

on behalf of Dr Ulas Tezel (Associate Editor) and R. Kerry Rowe (Subject Editor)
openscience@royalsociety.org

Comments to Author:

Reviewers' Comments to Author:

Reviewer: 1

Comments to the Author(s)

In this paper, effects of yeast addition on the biogas production performance of a food waste (FW) anaerobic digestion system were comprehensively investigated. Generally, the topic and the data analysis are significant, the results obtained would make a sound contribution to the fields of sludge treatment and anaerobic digestion. However, in the reviewer's opinion, there are some major and minor shortcomings needed further attention via a major revision, with the detail as follows:

1. What is the role of yeast in the anaerobic digestion? I find there are a contradictory statement about it in the introduction. Firstly, the author stated " the AD process with FW often results in the accumulation of volatile organic acids (VFAs) in the reaction system and the acidification reduces methane yield and destroys the stability of the digestive system" in Page 3, Line 17-20. Then in Page 4, Line 10-12, the authors stated " The addition of yeasts during anaerobic digestion can increase the acid production rate of anaerobic digestion and shorten the time of anaerobic digestion". Both statements obviously are contradictory. What exactly is the mechanism for enhanced digestion of FW led by yeast? It should be confirmed.
2. In this study, the yeast was added in twice. What is the reason? and how to chose the added time? Meanwhile, the characteristic of the yeast *Saccharomyces cerevisiae* should be provided.
3. Except for the yeast, some other pretreatment technologies are benefit to the anaerobic digestion of FW and they should be summarized and compared in the introduction. Some recent references maybe helpful, such as Reviews in Environmental Science and Bio-technology, 2019, 18(4):771-793(<https://doi.org/10.1007/s11157-019-09515-y>); Environmental Science and Pollution Research, 2019, 26(14): 13984- 13998 (<https://doi.org/10.1007/s11356-019-04798-8>)
4. In this study, the substrate also was added secondly. Why?
There are only one equation, so using 3.1 to label is improper.
5. Numerous grammar errors distributed across the paper, resulting in poor readability. I think the article in its current state is unable to be accepted unless the article will be re-written by some of the co-authors who are skillful in English.

for example:

Page 2, Line 4 " ...under decreased substrate conditions after biogas production was reduced was studied." ????

Page 2, Line 6-7, what do you mean " the recovery of gas production"?

Page 4, Line 8 " the effect and performance of direct yeast addition"??

Page 4, Line 19-20 " FW and sludge anaerobic fermentation."???

Reviewer: 2

Comments to the Author(s)

The authors must correct the grammar and choose proper words to make the manuscript more readable and ready for publication

Reviewer: 3

Comments to the Author(s)

Please find the attached file for review comments to enable you improve the quality of your manuscript.

Author's Response to Decision Letter for (RSOS-200443.R0)

See Appendix B.

Decision letter (RSOS-200443.R1)

Dear Dr Gao,

It is a pleasure to accept your manuscript entitled "Effect of yeast addition on the biogas production performance of a food waste anaerobic digestion system" in its current form for publication in Royal Society Open Science.

Best regards,

on behalf of Dr Ulas Tezel (Associate Editor) and R. Kerry Rowe (Subject Editor)
openscience@royalsociety.org

Appendix A

Manuscript title: **Effect of yeast addition on the biogas production performance of a food waste anaerobic digestion system**

Manuscript Number: RSOS-200443

General

This paper tried to address the “Effect of yeast addition on the biogas production performance of a food waste anaerobic digestion system”. The study is an interesting research aimed at achieving higher biodegradability and operational stability in an anaerobic digestion process of food waste supplemented with yeast. However, in order to improve the quality of this manuscript, the following revisions and amendments are recommended:

Abstract

1. The sentences in lines 3 and 4 of page 2 should be revised. Correct this sentence “*In the control group, the increase in the recovery of gas production was significantly reduced*” in lines 6 and 7, page 2.

Introduction

1. General language editing is required across the entire Introduction section, (line 11, page 3; line 16, 18, 19, page 4, etc).
2. Elaborate with more published articles on the mechanisms of yeast action on substrates and other operational factors during biogas production.

Materials and methods

1. Use more technical term not “domestication”.
2. For full disclosure, the source of the yeast and glucose should be stated (whether purchased or manufactured).
3. Is the AD reaction batch or continuous? Kindly state and was the AD reaction replicated? If yes, how many times? And then the digestion period is not stated.
4. Edit the statement in line 9,10 and 11, page 6.
5. Indicate how did the authors determine yeast and glucose dosages and addition intervals, any previous studies or experiments to support your decision.
6. What is the implication of the second yeast and glucose addition, is it economically viable if implemented on commercial scale?

Results and discussion

1. “... *fermentation were determined (Figure 2)*” in line 20, page 6, how was it “determined” or do you mean “recorded”?
2. Provide citation for “*Under standard conditions, 1 g of protein (calculated as VS) can produce 992 mL of biogas in anaerobic fermentation*” in line 10 and 11, page 7.

3. In section 3.1, there should be proper discussion of facts with adequate citation. Results should be stated earlier before discussions with literature information to avoid missing out salient discoveries in the study.
4. This study is not a review article; therefore, emphasis must be on the results of this study, then supporting literatures and such statements to justify the results.

Conclusion

1. Revise this conclusion section to reflect the essence of this study and the findings.
2. “*Especially, activated yeast was supplemented twice when AD process was inhibited with few daily biogas generation (nearly zero)*” this statement in line 20 and 21, page 11 could be misleading, while it may be true for your second addition, Fig. 2a denied the near zero yield statement for the first addition. Please revise.
3. Edit the language in line 21, page 11 and line 1, page 12.

Appendix B

Dear editors:

Thanks for your careful review with regard to our manuscript “Effect of yeast addition on the biogas production performance of a food waste anaerobic digestion system” (Manuscript ID: RSOS-200443).

These comments are very helpful for us to revise and improve our manuscript. We have made changes in the text based on the comments of two reviewers and also re-examined the manuscript. Revised parts are highlighted in **Yellow** in the paper. Point-to-point responses to the comments are listed as flowing (**Responses to reviewers’ questions in Blue words; Revisions in Red words**).

We appreciate for your consideration of our work, and hope that the responses and the revision could be acceptable.

Response to Reviewer 1

In this paper, effects of yeast addition on the biogas production performance of a food waste (FW) anaerobic digestion system were comprehensively investigated. Generally, the topic and the data analysis are significant, the results obtained would make a sound contribution to the fields of sludge treatment and anaerobic digestion. However, in the reviewer's opinion, there are some major and minor shortcomings needed further attention via a major revision, with the detail as follows:

Q1: What is the role of yeast in the anaerobic digestion? I find there are a contradictory statement about it in the introduction. Firstly, the author stated " the AD process with FW often results in the accumulation of volatile organic acids (VFAs) in the reaction system and the acidification reduces methane yield and destroys the stability of the digestive system" in Page 3, Line 17-20. Then in Page 4, Line 10-12, the authors stated " The addition of yeasts during anaerobic digestion can increase the acid production rate of anaerobic digestion and shorten the time of anaerobic digestion". Both statements obviously are contradictory. What exactly is the mechanism for enhanced digestion of FW led by yeast? It should be confirmed.

Replies: Thank you for your careful review and valuable comments of this paper.

The role of yeast in anaerobic digestion is that yeast can accelerate reaction rates in the hydrolysis stage of organic matter and promote the hydrolysis of food waste. And the addition of yeast promotes the hydrolysis of substrates and converts the degradable organic matter of substrates into neutral ethanol instead of acid propionic acid or butyric acid, which reduces the load of VFA in the system and provide more potentially available energy to methanogens, thereby improving the system's stability.

Revisions in revised manuscript (Page 4, Lines 18-23): It has been proved that yeast can accelerate reaction rates in the hydrolysis stage of organic matter and promote the hydrolysis of FW [21]. FW contains high enough levels of proteins, amino acids and trace elements from the hydrolysis and breakdown of food-stuffs, and can provide sufficient nutrition for the growth of yeast [22]. In the process of anaerobic digestion of food waste, yeast can convert the degradable organic matter of substrates into neutral ethanol instead of acid propionic acid or butyric acid, which reduces the load of VFA in the system [23, 24]. Through the addition of yeast to the FW substrate for ethanol pre-fermentation before AD, a large amount of ethanol is produced, and ethanol is gradually converted into acetic acid, which in turn can be easily used by methanogens, thereby increasing methane production, reducing the effect of acidification on the system, and enhancing the stability of AD[15, 25].

22. Ritchie RJ, Raghupathi SS. 2008 Al-toxicity studies in yeast using gallium as an aluminum analogue. *Biometals* 21, 379-393. (doi:10.1016/S0141-8130(98)00095-6)

23. Ma X, Yu M, Song N, Xu B, Gao M, Wu C, Wang Q. 2020 Effect of ethanol pre-fermentation on organic load rate and stability of semi-continuous anaerobic digestion of food waste. *Bioresource Technol.* 299, 122587. (doi:10.1016/j.biortech.2019.122587)

24. Mashair AS, Hongzhi M, Siyuan Y, Qunhui W, Maobing T. 2018 Concise review on ethanol production from food waste: development and sustainability. *Environmental Science & Pollution Research* 3. (doi:10.1007/s11356-018-2972-4)

Q2: In this study, the yeast was added in twice. What is the reason? and how to chose the added time? Meanwhile, the characteristic of the yeast *Saccharomyces cerevisiae* should be provided.

Replies: The purpose of adding yeast for the first time is to investigate the promotion effect and mechanism of yeast on anaerobic digestion of food waste. With the increase of anaerobic fermentation time, the daily biogas production of the two groups decreased significantly. Because the accumulation of volatile fatty acids in the system inhibits the activity of methanogens and further weakens their ability to metabolize VFAs, thus reducing the methane yield. In order to investigate the effect of yeast on reducing the load of VFAs, improving the system's stability and methane production, yeast was added on the 12th day of fermentation when the daily biogas production was significantly reduced.

The purpose of the second addition was to investigate whether the decrease of gas production to 0 was due to the depletion of substrate or the deactivation of methanogens. On the 37th day of fermentation, the daily biogas production of both experimental groups decreased to 0, so yeast and glucose were added for the second time on the 37th day of fermentation.

The yeast is Angel active yeast (produced by Angel Yeast Co., Ltd., Hubei, China), which is used as an additive to the anaerobic digestion system. Yeast contains 38-60% protein, 25% - 35% carbohydrate, 4-7% fat and 6-15% nucleic acid. The reason for adding yeast twice, adding time and the characteristic of the yeast are supplemented in the revised manuscript.

Revisions in revised manuscript (Page 6, Lines 2-5): In the yeast group, 2.0% (calculated as VS) of activated *Saccharomyces cerevisiae* was added when fermentation gas production had drastically reduced (day 12 of fermentation) to investigate the effect of yeast on reducing the load of VFAs, improving the system's stability and methane production.

(Page 6, Lines 8-10): In order to investigate whether the reduction of gas production to zero in the later stage of anaerobic fermentation is due to the depletion of substrate or the deactivation of methanogens, yeast and glucose were added for the second time.

(Page 5, Lines 17-19): Active dry yeast was produced by Angel Yeast Co., Ltd., Hubei,

China, which is used as an additive to the anaerobic digestion system. Yeast contains 38-60% protein, 25% - 35% carbohydrate, 4-7% fat and 6-15% nucleic acid. The glucose was produced by Aladdin Co., Ltd., Shanghai, China.

Q3: Except for the yeast, some other pretreatment technologies are benefit to the anaerobic digestion of FW and they should be summarized and compared in the introduction. Some recent references maybe helpful, such as Reviews in Environmental Science and Bio-technology, 2019, 18(4):771-793(<https://doi.org/10.1007/s11157-019-09515-y>); Environmental Science and Pollution Research, 2019, 26(14): 13984- 13998 (<https://doi.org/10.1007/s11356-019-04798-8>)

Replies: Thank you for your valuable advice, which is very helpful to our writing. We added the summary and comparison of pretreatment technology in the introduction of the revised manuscript (Page 3, Lines 21-23; Page 4, Lines 1-6).

Revisions in revised manuscript (Page 3, Lines 21-23; Page 4, Lines 1-6): Previous studies have identified several methods to enhance methane fermentation efficiency using FW substrate pretreatments, including mechanical, thermal, chemical, and biological pretreatments [12, 13]. Mechanical pretreatment enhances anaerobic process by increasing specific surface area. Li et al. studied the effect of heat pretreatment on the degradation of organics in FW, found that heat pretreatment increased the stagnation period of protein degradation (35-65%), and the cumulative gas production also increased linearly [14]. However, these methods require costly facilities and treatments. As an important part of biological pretreatment, ethanol pre-fermentation has become a popular research topic in recent years. Wu et al. found that ethanol pre-fermentation effectively alleviated the inhibition of acidification, greatly reduced the lag period, and significantly stimulated the growth of methanogens [15].

12. Yang Q, Wu B, Yao F, He L, Li X. 2019 Biogas production from anaerobic co-digestion of waste activated sludge: co-substrates and influencing parameters. Reviews in Environmental ence and Bio/Technology 18, 771-793. (doi:10.1007/s11157-019-09515-y)

13. Luo K, Pang Y, Yang Q, Wang D, Li X, Lei M, Huang Q. 2019 A critical review of volatile fatty acids produced from waste activated sludge: enhanced strategies and its applications. *Environ. Sci. Pollut. R.* 26, 13984-13998. (doi:10.1007/s11356-019-04798-8)

14. Li Y, Jin Y, Li J, Li H, Yu Z, Nie Y. 2017 Effects of thermal pretreatment on degradation kinetics of organics during kitchen waste anaerobic digestion. *Energy* 118, 377-386. (doi:10.1016/j.energy.2016.12.041)

15. Wu C, Wang Q, Yu M, Zhang X. 2015 Effect of ethanol pre-fermentation and inoculum-to-substrate ratio on methane yield from food waste and distillers' grains. *Appl. Energ.* 155, 846-853. (doi:10.1016/j.apenergy.2015.04.081)

Q4: In this study, the substrate also was added secondly. Why? There are only one equation, so using 3.1 to label is improper.

Replies: In this study, the substrate was added for the second time, because the daily biogas production in the later stage of fermentation has decreased to 0, which may be caused by the depletion of substrate in the anaerobic digestion system. And this paper wants to study whether adding yeast can restore biogas production when adding substrate after biogas production is reduced. The equation is corrected on Page 8, Line 7、 Line 12.

Original text (Page 8, Line 7、 Line 12): The calculation formula for protein biogas production is shown in equation (3.1).

Revisions in revised manuscript: The calculation formula for protein biogas production is shown in equation (1).

Q5: Numerous grammar errors distributed across the paper, resulting in poor readability. I think the article in its current state is unable to be accepted unless the article will be re-written by some of the co-authors who are skillful in English.

for example:

Page 2, Line 4 " ...under decreased substrate conditions after biogas production was reduced was studied." ????

Page 2, Line 6-7, what do you mean " the recovery of gas production"?

Page 4, Line 8 " the effect and performance of direct yeast addition"??

Page 4, Line 19-20 " FW and sludge anaerobic fermentation."???

Replies: We regret there were problems with the English. The paper has been carefully revised by a professional language editing service to improve the grammar and readability.

Original text (Page 2, Line 4): This work aimed to investigate the effect of adding yeast on biogas production performance under decreased substrate conditions after biogas production was reduced was studied.

Revisions in revised manuscript: This work aimed to investigate the effect of adding yeast on biogas production performance, **when substrate is added after biogas production is reduced.**

Original text (Page 2, Line 7): In the control group, the increase in the recovery of gas production was significantly reduced.

Revisions in revised manuscript: In the control group, **the increase of gas production was significantly reduced.**

Original text (Page 4, Line 16): However, presently, barely research works were performed to explore the effect and performance of direct yeast addition to an AD system of FW.

Revisions in revised manuscript (Page 5, Line 4): **However, there are few studies on adding yeast directly to the anaerobic digestion system of FW.**

Original text (Page 4, Lines 19-20): In this work, activated yeast was added to the fermentation broth of FW and sludge anaerobic fermentation.

Revisions in revised manuscript: In this work, **activated yeast was added to the fermentation broth of anaerobic fermentation of FW.**

Response to Reviewer 2

The authors must correct the grammar and choose proper words to make the

manuscript more readable and ready for publication

Replies or revisions in revised manuscript: Thanks for your approval and comments of this paper. I noticed that you reviewed this article very carefully and made a lot of detailed suggestions. I am very grateful for your work. You are a pretty serious and responsible reviewer, and it is my pleasure to be reviewed by you.

Comments for Abstract

Q1: The sentences in lines 3 and 4 of page 2 should be revised. Correct this sentence “In the control group, the increase in the recovery of gas production was significantly reduced” in lines 6 and 7, page 2.

Original text (page 2, Lines 3-4): This work aimed to investigate the effect of adding yeast on biogas production performance under decreased substrate conditions after biogas production was reduced was studied.

Replies: Thanks for your comment. We have made changes on the Lines 3-4 of page 2 of the revised text.

Revisions in revised manuscript: This work aimed to investigate the effect of adding yeast on biogas production performance, **when substrate is added after biogas production is reduced.**

Original text (Page 2, Lines6-7): In the control group, the increase in the recovery of gas production was significantly reduced.

Replies: Thanks for your comment. We have made changes on the Lines 6-7 of page 2 of the revised text.

Revisions in revised manuscript: In the control group, **the increase of gas production was significantly reduced.**

Comments for Introduction

Q2: General language editing is required across the entire Introduction section, (line 11, page 3; line 16, 18, 19, page 4, etc).

Replies: Thanks for your comment. We have edited the introduction of the revised manuscript.

Original text (Page 3, Lines 2-4): Especially, in China, the proportion of FW is 30%–60% compared with 23% in Japan, 15%–25% in Europe and only 12% in the United States.

Revisions in revised manuscript (Page 3, Lines 2-3): The proportion is 30% - 60% in China, while it is 23% in Japan, 15% - 25% in Europe, and only 12% in the United States.

Original text (page 3, Line 5): Improper handling of FW would cause problems in the fields of food safety and environmental pollution.

Revisions in revised manuscript: Improper handling of FW would cause problems of food safety and environmental pollution.

Original text (page 3, Lines 5-6): However, the approach of supplementing iron would take a long time to relieve acidification and restore methane production, which exhibited low efficiency.

Revisions in revised manuscript: However, the approach of supplementing zero valent iron would take a long time to relieve acidification and restore methane production, which caused low efficiency.

Original text (Page 3, Line11): On the other hand, biogas produced from AD is easily to be separated and utilised in engineering. Further, AD owns simple process flow, high reactor efficiency, low pollution load and , high economic benefits, which even the highest potential for FW recycling.

Revisions in revised manuscript: On the other hand, biogas produced from AD is easily separated and utilised in applications. Further, AD has the advantages of simple process flow, high reactor efficiency, low pollution load, high economic benefits, and even the highest potential for FW recycling.

Original text (Page 4, Line 16): However, presently, barely research works were performed to explore the effect and performance of direct yeast addition to an AD system of FW.

Revisions in revised manuscript: However, there are few studies on adding yeast directly to the anaerobic digestion system of FW.

Original text (Page 4, Lines 19-20): In this work, activated yeast was added to the fermentation broth of FW and sludge anaerobic fermentation.

Revisions in revised manuscript: In this work, **activated yeast was added to the fermentation broth of anaerobic fermentation of FW.**

Q3: Elaborate with more published articles on the mechanisms of yeast action on substrates and other operational factors during biogas production.

Replies: Thank you for your valuable advice, which is very helpful to our writing. In the introduction of the revised manuscript, we quoted more published articles, and elaborated the mechanism of yeast action on substrates and other operational factors during biogas production. (Page 3, Lines 21-23; Page 4, Lines 1-6).

Revisions in revised manuscript (Page 4, Lines 16-23; Page 5, Lines 1-4): Yeast has the characteristics of osmotic pressure resistance, acid resistance and high metabolic efficiency, which is widely used in the treatment of high-concentration organic wastewater, including the treatment of toxic and refractory pollutant-containing wastewater[20]. It was suggested that yeast can accelerate reaction rates in the hydrolysis stage of organic matter and promote the hydrolysis of FW [21]. **FW contains high enough levels of proteins, amino acids and trace elements from the hydrolysis and breakdown of food-stuffs, and can provide sufficient nutrition for the growth of yeast [22]. In the process of anaerobic digestion of food waste, yeast can convert the degradable organic matter of substrates into neutral ethanol instead of acid propionic acid or butyric acid, which reduces the load of VFA in the system [23, 24].** Through the addition of yeast to the FW substrate for ethanol pre-fermentation before AD, a large amount of ethanol is produced, and ethanol is gradually converted into acetic acid, which in turn can be easily used by methanogens, thereby increasing methane production, reducing the effect of acidification on the system, and enhancing the stability of AD[15, 25].

22. Ritchie RJ, Raghupathi SS. 2008 Al-toxicity studies in yeast using gallium as an aluminum analogue. *Biometals* 21, 379-393. (doi:10.1016/S0141-8130(98)00095-6)

23. Ma X, Yu M, Song N, Xu B, Gao M, Wu C, Wang Q. 2020 Effect of ethanol pre-fermentation on organic load rate and stability of semi-continuous anaerobic digestion of food waste. *Bioresource Technol.* 299, 122587.

(doi:10.1016/j.biortech.2019.122587)

24. Mashair AS, Hongzhi M, Siyuan Y, Qunhui W, Maobing T. 2018 Concise review on ethanol production from food waste: development and sustainability. *Environmental Science & Pollution Research* 3. (doi:10.1007/s11356-018-2972-4)

Comments for Materials and methods

Q4: Use more technical term not “domestication”.

Replies: Thanks for your comment. We have made changes on the Page 5, Line 5 of the revised manuscript.

Original text (Page 5, Line 5): The inoculated sludge was taken from a rural biogas station in Beijing's Pinggu District and used after domestication.

Revisions in revised manuscript: The inoculated sludge was taken from a rural biogas station in Beijing's Pinggu District and used after **acclimation**.

Q5: For full disclosure, the source of the yeast and glucose should be stated (whether purchased or manufactured).

Replies: Thanks for your comment. The source of the yeast and glucose and the characteristic of the yeast are supplemented in the revised manuscript.

Revisions in revised manuscript (Page 5, Lines 17-19): **Active dry yeast was produced by Angel Yeast Co., Ltd., Hubei, China**, which is used as an additive to the anaerobic digestion system. **Yeast contains 38-60% protein, 25% - 35% carbohydrate, 4-7% fat and 6-15% nucleic acid. The glucose was produced by Aladdin Co., Ltd., Shanghai, China.**

Q6: Is the AD reaction batch or continuous? Kindly state and was the AD reaction replicated? If yes, how many times? And then the digestion period is not stated.

Replies: Thanks for your comment. Anaerobic digestion is a batch reaction. We have made changes on the Page 5, Line 23 of the revised manuscript. The anaerobic digestion was not repeated in this experiment, and the digestion time was 50 days.

Revisions in revised manuscript (Page 5, Line 23): **In this study, two sets of experiments**

were conducted in batch mode. The AD reactions were carried out in 500-mL anaerobic bottles with 400-mL working volume.

(Page 6, Line 12): The experiments lasted for 50 days. Sampling was withdrawn and various parameters were determined at regular intervals. Biogas production was recorded daily.

Q7: Edit the statement in line 9,10 and 11, page 6.

Replies: Thanks for your comment. We have made changes on the Page 6, Lines 9-11 of the revised manuscript.

Original text (Page 6, Lines 9-11): A Shimadzu CP3800 gas chromatograph equipped with a DB-FFAP capillary column (30 m × 0.53 mm × 0.5 μm, Agilent Technologies Co., Ltd.) was used to determine the samples. The biogas was collected by the drainage method with alkaline solution.

Revisions in revised manuscript: Then, the samples were determined on a Shimadzu CP3800 gas chromatograph equipped with a DB-FFAP capillary column (30 m × 0.53 mm × 0.5 μm, Agilent Technologies Co., Ltd.) and a flame ionisation detector. The biogas was collected by the drainage method with alkaline.

Q8: Indicate how did the authors determine yeast and glucose dosages and addition intervals, any previous studies or experiments to support your decision.

Replies: Thanks for your comment. We have researched the effect of ethanol pre-fermentation on organic load rate and stability of anaerobic digestion systems and published these results in other papers [1-3]. In these studies, we determined the dosage of yeast and glucose and the activation method of yeast. The yeast was first inoculated into 100 mL sterilized growth medium (2% glucose) with a mass ratio of 2% (W/V medium) in a 250 mL flask for activity recovery. This pre-cultivation was performed anaerobically at 35 °C for 2 h with mixing at 150 rpm using a rotary shaker.

The addition time of yeast and glucose is selected according to the situation of anaerobic fermentation system. In order to investigate the effect of yeast on reducing the load of VFAs, improving the system's stability and methane production, yeast was

added on the 12th day of fermentation when the daily biogas production was significantly reduced. On the 37th day of fermentation, the daily biogas production of both experimental groups decreased to 0, so yeast and glucose were added for the second time on the 37th day of fermentation. We have supplemented this part in the revised manuscript (Page 6, Lines 3-7; Lines 9-11).

1. Wu C, Wang Q, Yu M, Zhang X. 2015 Effect of ethanol pre-fermentation and inoculum-to-substrate ratio on methane yield from food waste and distillers' grains. *Appl. Energ.* 155, 846-853. (doi:10.1016/j.apenergy.2015.04.081)

2. Yu M, Wu C, Wang Q, Sun X, Ren Y, Li Y. 2018 Ethanol prefermentation of food waste in sequencing batch methane fermentation for improved buffering capacity and microbial community analysis. *Bioresour. Technol.* 248, 187-193. (doi:10.1016/j.biortech.2017.07.013)

3. Ma X, Yu M, Song N, Xu B, Gao M, Wu C, Wang Q. 2020 Effect of ethanol pre-fermentation on organic load rate and stability of semi-continuous anaerobic digestion of food waste. *Bioresour. Technol.* 299, 122587. (doi:10.1016/j.biortech.2019.122587)

Revisions in revised manuscript (Page 6, Lines 6-7): The method for the yeast addition and activation has been mentioned in previous research [15].

15. Wu C, Wang Q, Yu M, Zhang X. 2015 Effect of ethanol pre-fermentation and inoculum-to-substrate ratio on methane yield from food waste and distillers' grains. *Appl. Energ.* 155, 846-853. (doi:10.1016/j.apenergy.2015.04.081)

(Page 6, Lines 3-6): In the yeast group, 2.0% (calculated as VS) of activated *Saccharomyces cerevisiae* was added when fermentation gas production had drastically reduced (day 12 of fermentation) to investigate the effect of yeast on reducing the load of VFAs, improving the system's stability and methane production. The method for the yeast addition and activation has been mentioned in previous research [15].

(Page 6, Lines 9-11): In order to investigate whether the reduction of gas production to zero in the later stage of anaerobic fermentation is due to the depletion of substrate or the deactivation of methanogens, yeast and glucose were added for the

second time (No biogas generation, 37 d) .

Q9: What is the implication of the second yeast and glucose addition, is it economically viable if implemented on commercial scale?

Replies: Thanks for your comment. The purpose of the second addition was to investigate whether the decrease of gas production to 0 was due to the depletion of substrate or the deactivation of methanogens. On the 37th day of fermentation, the daily biogas production of both experimental groups decreased to 0, so yeast and glucose were added for the second time on the 37th day of fermentation. We have supplemented this part in the revised manuscript (Page 6, Lines 9-11). According to our research results, after the second addition of yeast and glucose, the daily biogas production is still significantly increased, which indicates that the addition of yeast is conducive to the rapid recovery of biogas production in the anaerobic system. Therefore, if it is implemented on a commercial scale, it is economically feasible when the substrate in the anaerobic digestion system is not exhausted and the methane bacteria are not inactivated.

Revisions in revised manuscript (Page 6, Lines 9-11): In order to investigate whether the reduction of gas production to zero in the later stage of anaerobic fermentation is due to the depletion of substrate or the deactivation of methanogens, yeast and glucose were added for the second time.

Comments for Results and discussion

Q10: "... fermentation were determined (Figure 2)" in line 20, page 6, how was it "determined" or do you mean "recorded"?

Replies: Thanks for your comment. We have made changes on the Page 7, Line19 of the revised manuscript.

Original text (Page 6, Line 20): The time course of daily biogas production and cumulative biogas production of two groups within anaerobic fermentation were determined (Figure 2).

Revisions in revised manuscript(Page 7, Line19): The time course of daily biogas

production and cumulative biogas production of two groups within anaerobic fermentation were recorded (Figure 2).

Q11: Provide citation for “Under standard conditions, 1 g of protein (calculated as VS) can produce 992 mL of biogas in anaerobic fermentation” in line 10 and 11, page 7.

Replies: Thanks for your comment. We have made changes on the Page 8, Lines 9-10 of the revised manuscript.

Original text (Page 7, Lines 10-11): Under standard conditions, 1 g of protein (calculated as VS) can produce 992 mL of biogas in anaerobic fermentation.

Revisions in revised manuscript (Page 8, Lines 9-10): Under standard conditions, 1 g of protein (calculated as VS) can produce 496 mL of biogas in anaerobic fermentation [26]. The calculation formula for protein biogas production is shown in equation (1). The average protein content in yeast was 50%, and the amount of biogas produced by adding yeast itself was 248 mL. Therefore, deducting $4.6 \text{ mL} \cdot \text{g}^{-1} \cdot \text{VS}^{-1}$ of the biogas produced by the yeast itself in the yeast group revealed that biogas production by the yeast group had increased by 33.2% compared with that by the control group.

26. Ller HBM, Sommer SG, Ahring BK. 2004 Methane productivity of manure, straw and solid fractions of manure. *Biomass Bioenerg.* **26**, 485-495. (doi:10.1016/j.biombioe.2003.08.008)

Q12: In section 3.1, there should be proper discussion of facts with adequate citation. Results should be stated earlier before discussions with literature information to avoid missing out salient discoveries in the study.

Replies: Thank you for your valuable advice, which is very helpful to our writing. In Section 3.1 of the revised manuscript, we had a supplementary discussion of facts with adequate citation (Page 8, Lines 8-21).

Revisions in revised manuscript (Page 8, Lines 8-21): The addition of activated yeast can drastically promote the anaerobic digestion of methanogenesis in FW. In other study, yeast was added to the substrate for ethanol prefermentation in the sequencing

batch methane fermentation of food waste. The results showed that methane production in the EP group (254 mL/g VS) was higher than in the control group (35 mL/g VS) [25]. It was consistent with the results of this study.

This effect may be attributed to the reduction in the production of VFAs by the addition of activated yeast. Because of the glycolytic metabolic pathway in *S. cerevisiae*, the end products are mainly ethanol and acetic acid after yeast addition [27]. Moreover, the acidification in AD system is considered as the problematic inhibition of biogas production. Thus, FW substrate was mainly metabolised into ethanol by additional yeast, instead of VFAs. The ethanol could be gradually converted into acetic acid, which can be easily used by methanogens, thereby increasing methane production and enabling the stable operation of anaerobic fermentation[4]. And the addition of yeast can improve the relative abundance of methane-producing bacteria in anaerobic digestion system [25].

25. Yu M, Wu C, Wang Q, Sun X, Ren Y, Li Y. 2018 Ethanol prefermentation of food waste in sequencing batch methane fermentation for improved buffering capacity and microbial community analysis. *Bioresource Technol.* **248**, 187-193. (doi:10.1016/j.biortech.2017.07.013)

27. Shen Y, Zhao XQ, Ge XM, Bai FW. 2009 Metabolic flux and cell cycle analysis indicating new mechanism underlying process oscillation in continuous ethanol fermentation with *Saccharomyces cerevisiae* under VHG conditions. **27**, 1118-1123. (doi:10.1016/j.biotechadv.2009.05.013)

Q13: This study is not a review article; therefore, emphasis must be on the results of this study, then supporting literatures and such statements to justify the results.

Replies: Thank you for your valuable advice, which is very helpful to our writing. We have revised sections 3.2 and 3.3 of the revised manuscript. After the revision, we first emphasized the results of this study, and then provided literatures and statements

to prove the rationality of the results.

Revisions in revised manuscript (Page 9, Lines 5-16): As shown in Figure 3-a, the pH values of the two experimental groups gradually decreased but were never less than 6.5. Therefore, acidification did not occur during anaerobic fermentation. This phenomenon also explains why the daily output of biogas in the early stage of fermentation in Figure 2-a did not decrease to zero, even if it decreased. After 8 days of anaerobic fermentation, the pH of the two groups of experiments rapidly increased probably because of the buffering effect of the system itself. After the first addition of yeast or glucose, the pH value of the yeast group did not significantly differ from that of the control group, which was above 7.5. According to current research reports, due to the differences in the nature of the additives, the pH of a methane fermentation system that can maintain stable operation is in the range of 6.5–8.2[30]. The pH value also plays a vital role in regulating the activities of microorganisms. The optimum pH range of acid-producing bacteria is in the range of 4.0–8.5; the limit of the pH value of methanogenic bacteria is in the range of 6.5–7.2[31].

(Page 10, Lines 7-20): Figure 3-d shows the variation in TVFA concentration to TA (TVFA/TA) values with fermentation time in the two groups of experiments. After the addition of activated yeast on day 12 of fermentation, the alkalinity of the system increased rapidly (Figure 3-b). The TVFA/TA ratio continued to decrease, and the yeast system began to produce a large amount of gas. The TVFA/TA value of the yeast group was lower than that of the control group, indicating that the addition of yeast to the anaerobic digestion system can adjust alkalinity and reduce TVFA concentration, thereby improving the stability of anaerobic digestion. The TVFA/TA values of the two groups of experiments never exceeded 0.4. This result also proved that the system was not acidified, and the reduction in gas production before the addition of yeast or glucose may be caused by substrate exhaustion. Similar findings have been found in previous studies. The ratio of TVFA/TA can be used as an early warning of digestive system imbalance. TVFA/TA shows the ratio between compounds that causes a decrease in pH and compounds that maintain alkalinity in the system. This index sensitively reflects the ability of the anaerobic digestion system to withstand acidification. When TVFA/TA

exceeds 0.4, the acidification of the anaerobic system is about to be destabilised. When TVFA/TA exceeds 0.6, the acidification of the anaerobic system will be completely imbalanced[23]. Comparing the two groups' parameters, such as pH, alkalinity, TVFA concentration and TVFA/TA ratio, revealed that these parameters followed the same trend. However, the alkalinity of the yeast group was slightly higher than that of the control group, the TVFA and TVFA/TA of the yeast group were slightly lower, thereby indicating that the addition of activated yeast improved the stability of anaerobic digestion.

(Page 11, Lines 3-23): The main VFAs components detected in the two groups were acetic, propionic and butyric acids (Figures 4-a and 4-b). Valeric acid, isovaleric acid and iso-butyric acid were also detected at low concentrations. The acetic acid concentration of the yeast group reached a maximum of 7.97 g·L⁻¹ on day 8, and then began to decrease with the consumption of butyric acid. From days 11 to 15, the concentration of butyric acid decreased rapidly. Daily biogas production also increased (Figure 2-a). Propionic acid gradually accumulated with the AD progression of fermentation, thereby propionic acid-type fermentation was considered to be established. In anaerobic fermentation, VFAs containing more than two carbon chains cannot be directly used as a substrate by methanogens, and thus, which are easily accumulated during fermentation [3]. Propionic acid is a common short-chain fatty acid. From the perspective of metabolism, propionic acid is usually converted into acetic acid and hydrogen. However, acetic acid generation by propionic acid and butyric acid is an endothermic reaction that is difficult to perform thermodynamically, and the conversion of propionic acid to acetic acid is difficult. It was suggested that propionic acid is a disadvantageous substrate for microorganisms [19]. Therefore, when propionic acid-type fermentation is occurred, the utilisation of organic acids in the methanogenic phase is inhibited and acids accumulation is promoted, which adversely affects methanogenesis. It was found that high temperature and alkaline pH are favourable for propionic acid generation [34]. On the other hand, the yield of VFAs, especially propionic acid, was also suggested to be enhanced by adjusting the pH level. Further, the key enzyme activity associated with propionic acid formation was the highest, when

the system pH was set at 8.0[3]. After 12 days of fermentation, the propionic acid concentration of the yeast group was lower than that of the control group, indicating that the addition of yeast can play a certain preventive role when propionic acid fermentation has occurred in the anaerobic fermentation system.

Comments for Conclusion

Q14: Revise this conclusion section to reflect the essence of this study and the findings.

Replies: Thank you for your valuable advice, which is very helpful to our writing. We have revised conclusion section of the revised manuscript.

Revisions in revised manuscript (Page 12, Lines 15-21): The effect of yeast addition on AD of FW was investigated in this study. The results showed that the addition of yeast can restore and promote the biogas production by anaerobic digestion. Moreover, AD with yeast addition exhibited a high VFAs consumption rate and low propionic acid concentration, which prevented the excessive acidification phenomenon. By adding yeast, FW was converted into ethanol as a slow-release substrate, instead of VFAs. The ethanol could be gradually converted into acetic acid, which can be easily used by methanogens, thereby increasing methane production and enabling the stable operation. Therefore, yeast addition was suggested as a feasible approach to maintain a stable AD system.

Q15: “Especially, activated yeast was supplemented twice when AD process was inhibited with few daily biogas generation (nearly zero)” this statement in line 20 and 21, page 11 could be misleading, while it may be true for your second addition, Fig. 2a denied the near zero yield statement for the first addition. Please revise.

Replies: Thanks for your comment. There is a real ambiguity in this sentence. And this sentence belongs to the description of experimental, so it was deleted in the conclusion part.

Q16: Edit the language in line 21, page 11 and line 1, page 12.

Replies: Thanks for your comment. We have made changes on the Page 12, Lines

16-18 of the revised manuscript.

Original text (Page 11, Line 21): Yeast addition was found to recovery and further promote the performance of methane production in terms of cumulative biogas production and the highest daily biogas generation.

Revisions in revised manuscript (Page 12, Line 16): The results showed that the addition of yeast can restore and promote the biogas production by anaerobic digestion.

Original text (Page 11, Line 21): FW substrate was mainly metallised into ethanol as a slow-release matrix by additional yeast, instead of VFAs.

Revisions in revised manuscript (Page 12, Lines 18): By adding yeast, FW was converted into ethanol as a slow-release substrate, instead of VFAs.

Response to Reviewer 3

Please find the attached file for review comments to enable you improve the quality of your manuscript.

Replies: Thank you for your careful reading of the manuscript. Your suggestions on revision are of great help to us. You are a pretty serious and responsible reviewer, and it is our pleasure to be reviewed by you. We carefully read the attached file for review comments and made the following modifications to the revised manuscript.

Q1: The authors should describe briefly the experiment to make it more understandable (Page 2, Line 5).

Original text: The daily biogas production of the yeast group after the first and second additions increased by 520 and 550 mL, respectively, and the gas production was relatively stable.

Replies: Thanks for your comment. We have made changes on the Page 2, Lines 4-6 of the revised manuscript.

Revisions in revised manuscript (Page 2, Lines 4-6): The results showed that the daily biogas production increased 520 ml and 550 ml by adding 2.0% (VS) of activated yeast on the 12th and 37th day of anaerobic digestion, respectively, and the gas production was relatively stable. **Q2:** What is the control group, more clarified statment is needed

(Page 2, Line 6).

Original text: In the control group, the increase in the recovery of gas production was significantly reduced.

Replies: Thanks for your comment. We have made changes on the Page 2, Line 7 of the revised manuscript.

Revisions in revised manuscript (Page 2, Line 7): In the control group without yeast, the increase of gas production was significantly reduced.

Q3: by reading this abstract, the readers should know what the second addition is.

Original text (Page 2, Line 7): After the second addition of substrate, the increase was only 60 mL more than the production before the addition.

Replies: Thanks for your comment. We have made changes on the Page 2, Lines 4-7 of the revised manuscript.

Revisions in revised manuscript (Page 2, Lines 4-7): The results showed that the daily biogas production increased 520 ml and 550 ml by adding 2.0% (VS) of activated yeast on the 12th and 37th day of anaerobic digestion, respectively, and the gas production was relatively stable.

After the second addition of substrate and yeast, biogas production only increased 60 ml compared with that before the addition.

Q4: This sentence is grammatically incorrect (Page 3, Lines 2-4).

Original text (Page 3, Lines 2-4): Especially, in China, the proportion of FW is 30%–60% compared with 23% in Japan, 15%–25% in Europe and only 12% in the United States.

Replies: Thanks for your comment. We have made changes on the Page 2, Lines 4-7 of the revised manuscript.

Revisions in revised manuscript (Page 3, Lines 2-3): The proportion is 30% - 60% in China, while it is 23% in Japan, 15% - 25% in Europe, and only 12% in the United States.

Q5: This could cause confusion (Page 8, Line 14).

Original text (Page 8, Line 14): the limit of the pH value of methanogenic bacteria is in the range of 6.5–7.2.

Replies: Thanks for your comment. We have made changes on the Page 2, Lines 4-7 of the revised manuscript.

Revisions in revised manuscript (Page 9, Line 14): The optimum pH range of methanogenic bacteria is in the range of 6.5–7.2.

Q6: You have pointed out the grammatical mistakes, misuses of words, and unclear descriptions.

Replies: Thanks for your comment. We have made changes in the revised manuscript, which were listed below.

(1) Original text (page 2, Lines 3-4): This work aimed to investigate the effect of adding yeast on biogas production performance under decreased substrate conditions after biogas production was reduced was studied.

Revisions in revised manuscript: This work aimed to investigate the effect of adding yeast on biogas production performance, when substrate is added after biogas production is reduced.

(2) Original text (page 3, Line 5): Improper handling of FW would cause problems in the fields of food safety and environmental pollution.

Revisions in revised manuscript: Improper handling of FW would cause problems of food safety and environmental pollution.

(3) Original text (page 3, Line 11): On the other hand, biogas produced from AD is easily to be separated and utilised in engineering.

Revisions in revised manuscript: On the other hand, biogas produced from AD is easily separated and utilised in applications.

(4) Original text (page 3, Lines 5-6): However, the approach of supplementing iron would take a long time to relieve acidification and restore methane production, which exhibited low efficiency.

Revisions in revised manuscript: However, the approach of supplementing zero valent iron would take a long time to relieve acidification and restore methane

production, which **caused** low efficiency.

(5) Original text (page 5,Line 5): The FW used in the experiments were collected from the student canteen of the University of Science and Technology Beijing, and bones and peels were sorted from the rest of the waste.

Revisions in revised manuscript: The FW used in the experiments were collected from the student canteen of the University of Science and Technology Beijing. **Large bones and pericarps of FW were first removed. The remaining waste was shredded and then stored at 20°C.**

(6) Original text (page 6,Line 11): The biogas was collected by the drainage method with alkaline solution.

Revisions in revised manuscript: **The biogas was collected by the drainage method with alkaline.**

(7) Original text (page 7,Line 22): Thus, FW substrate was mainly metallised into ethanol by additional yeast, instead of VFAs.

Revisions in revised manuscript: Thus, FW substrate was mainly **converted into** ethanol by additional yeast, instead of VFAs.

(8) Original text (page 9,Lines 3-4): The added yeast may compete with the substrate to promote the production of neutral ethanol, reduce the amount of VFAs and reduce the consumption of alkalinity in the system.

Revisions in revised manuscript(page 9,Lines 21-22): **The addition of yeast may promote the hydrolysis of organics to produce neutral ethanol,** reduce the amount of VFAs and reduce the consumption of alkalinity in the system.

(9) Original text (page 10, Line 11): However, acetic acid generation by propionic acid and butyric acid is an endothermic reaction that is difficult to perform thermodynamically.

Revisions in revised manuscript(page 11, Line12): However, acetic acid generation by propionic acid and butyric acid is an endothermic reaction that is difficult to **occur** thermodynamically.

(10) Original text (page 11, Lines 13-14): Zhao et al. appointed that the addition of yeast can increase the production of ethanol and reduce the production of lactic,

propionic and butyric acids.

Revisions in revised manuscript(page 12, Lines 8-9): The research results of Zhao et al. showed that the addition of yeast can increase the production of ethanol and reduce the production of lactic, propionic and butyric acids.